# Visible and Infrared Image Fusion of Forest Fire Scenes Based on Generative Adversarial Networks with Multi-Classification and Multi-Level Constraints

Qi Jin, Sanqing Tan *, Gui Zhang, Zhigao Yang, Yijun Wen, Huashun Xiao and Xin Wu

College of Forestry, Central South University of Forestry and Technology, Changsha 410004, China; j651168671@163.com (Q.J.); csfu3s@163.com (G.Z.); zgyang@126.com (Z.Y.); m19174989772@163.com (Y.W.); hsxiao@126.com (H.X.); 18711751279@163.com (X.W.)
* Correspondence: csuft0002@163.com

**Abstract:** Aimed at addressing deficiencies in existing image fusion methods, this paper proposed a multi-level and multi-classification generative adversarial network (GAN)-based method (MMGAN) for fusing visible and infrared images of forest fire scenes (the surroundings of firefighters), which solves the problem that GANs tend to ignore visible contrast ratio information and detailed infrared texture information. The study was based on real-time visible and infrared image data acquired by visible and infrared binocular cameras on forest firefighters' helmets. We improved the GAN by, on the one hand, splitting the input channels of the generator into gradient and contrast ratio paths, increasing the depth of convolutional layers, and improving the extraction capability of shallow networks. On the other hand, we designed a discriminator using a multi-classification constraint structure and trained it against the generator in a continuous and adversarial manner to supervise the generator, generating better-quality fused images. Our results indicated that compared to mainstream infrared and visible image fusion methods, including anisotropic diffusion fusion (ADF), guided filtering fusion (GFF), convolutional neural networks (CNN), FusionGAN, and dual-discriminator conditional GAN (DDcGAN), the MMGAN model was overall optimal and had the best visual effect when applied to image fusions of forest fire surroundings. Five of the six objective metrics were optimal, and one ranked second-to-optimal. The image fusion speed was more than five times faster than that of the other methods. The MMGAN model significantly improved the quality of fused images of forest fire scenes, preserved the contrast ratio information of visible images and the detailed texture information of infrared images of forest fire scenes, and could accurately reflect information on forest fire scene surroundings.

**Keywords:** forest fire scenes; visible and infrared images; image fusion; deep learning; GAN

## 1. Introduction

Forest fire surroundings are affected by terrain, meteorology, light, smoke, fire spread, and other factors, making it difficult to judge the internal conditions of the forest fire scene, which can easily lead to casualties if rescuers make errors in judgment during operations [1]. Therefore, firefighters need to quickly and accurately understand the conditions inside the fire and make avoidance decisions. Forest fire surroundings have obvious visible and infrared image feature information. Visible images of forest fire scenes can clearly express texture detail information, but they are difficult to adapt to changes in forest fire surroundings; infrared images of forest fire scenes are less affected by light and smoke and can better reflect the changes in forest fire surroundings, but it is difficult to reflect texture detail features in them.

Due to the complexity of forest fire scene surroundings, relying only on single-sensor data often does not provide enough information to support firefighters' decision-making. Image fusion technology has important advantages in forest fire scene rescue. By fusing

visible and infrared images of forest fires, the advantageous features of both sources can be fully utilized to provide more comprehensive and accurate fire scene information. Visible light images can provide clear texture details to help firefighters recognize objects and terrain, but they are affected by factors such as light and smoke. Infrared images, on the other hand, can better reflect the distribution of hot spots and fire expansion, and are relatively insensitive to light and smoke. Therefore, the study of visible and infrared image fusion in the scene of forest fires can make up for the limitations of each of the images, provide more comprehensive and accurate information about the fire scene, improve the efficiency and safety of rescuers, and thus significantly reduce the casualties caused by forest fires. Image fusion technology has gained widespread attention for its convenient, fast and economical features, assisting firefighters in making accurate decisions and providing an efficient method for fire rescue. At the same time, the application of image fusion technology can also provide a reference and inspiration for image processing and analysis in other fields and promote the development and application of imaging technology.

In addition, many studies have shown that image fusion techniques have the potential for practical applications in forest fire scene rescue. For example, Dios et al. used image fusion techniques to improve the accuracy of fire source localization, fire monitoring, and fire boundary identification [2]. Nemalidinne et al. used visible and infrared image fusion techniques to improve fire detection [3]. These findings suggest that image fusion techniques have great potential to improve the efficiency and safety of firefighters in forest fire rescues.

The scene of forest fire image fusion is a special form of image fusion field. Current visible and infrared image fusion methods can be classified into traditional methods and deep learning-based methods according to their principles. Traditional image fusion methods mainly include spatial domain methods and transform domain methods. Representative methods include logical filtering methods [4], color space methods [5], multi-scale transform methods [6], principal component analysis methods [7], sparse representation methods [8], and wavelet transform fusion methods [9]. However, traditional fusion methods have large limitations in terms of feature extraction and fusion rules. On the one hand, the diversity of source images makes the manual design of feature extraction and fusion rules increasingly complex; on the other hand, the generalizability of traditional methods is limited, and further improvement of fusion performance faces difficulties.

In recent years, the rapid development of deep learning has driven great progress in the field of image fusion. Deep learning-based methods utilize the powerful nonlinear fitting ability of neural networks to make fused images with desired distributions and further improve the performance of fused images. Convolutional neural networks (CNN) construct deep structural neural networks by training on natural samples, which enables a deeper image feature extraction to maintain the integrity of structural information and the preservation of detailed information in fused images [10]. CNNs have achieved good performance in multifocal images, but their computation is time-consuming, and the fusion performance depends on the characteristics of the training samples [11]. Ma et al. innovatively introduced generative adversarial networks (GAN) into the field of image fusion, combining adversarial learning and content-specific loss bootstrapping to achieve the preservation of significant contrast and texture details in fused images in an unsupervised situation [12]. However, the aforementioned methods do not sufficiently consider the extraction of information from shallow networks when extracting source image information, so there is still room for further improvement in fusion performance. If we can improve the degree of information utilization in the training process of GANs and achieve a good balance in maintaining visible and infrared image information, we are bound to further improve the performance of image fusion.

The superior performance of existing deep learning-based methods relies on a large number of labeled (reference image) datasets. However, labeled images are difficult to obtain for forest fire scene image fusion tasks. In order to make full use of the small amount of labeled data and a large amount of unsupervised data, this paper proposes the

visible and infrared image fusion of forest fire scenes using a method based on a multi-level and multi-classification GAN, referred to as MMGAN, which divides the input of the generator into a gradient path and intensity path, constructs primary and secondary information loss functions, and improves the depth of the convolutional layer of the generator, enhancing the information extraction capability of the shallow network. A multi-classifier is used in the discriminator, which determines the probability that the input is a visible image and an infrared image. By constraining these two probabilities through successive standoff learning, the generator can align the probability distributions of both IR and visible images. This effectively prevents the dilution of dominant information, resulting in fused images with pronounced contrast and intricate texture details. The comparative experimental results show that the proposed method can better accomplish the fusion of effective information from visible and infrared images of the forest fire scene, and the proposed method has better fusion performance compared with other, similar fusion methods. In addition, the proposed method has good generalizability and can be extended to any visible and infrared image fusion dataset.

## 2. Materials and Methods

### 2.1. Data and Pre-Processing

The visible and infrared image data of the forest fire scene in this study were acquired in real time from forest fires in Xintian County, Yongzhou City, Hunan Province, through a visible and infrared binocular camera integrated into a single firefighter's helmet. Xintian County is located in the southern part of Hunan Province, a region rich in forest resources and characterized geographically by mountainous, hilly, and forest-covered areas. The forest fire encompasses multiple ignition points and areas, yielding a wealth of visible and infrared image data. The acquired image data captured the characteristics of the fire smoke, burned areas and surroundings, and recorded the different stages and locations of the fire, providing a valuable data resource for research.

The TNO dataset comprises multispectral images of various scenes captured by different multi-band camera systems, including Athena, DHV, FEL and TRICLOBS. It is a widely used public research dataset in the field of infrared and visible image fusion, and its images have been subjected to strict image alignment [13].

In this paper, the forest fire scene's visible and infrared images and the TNO public dataset were selected as the data for the comparison experiments. The number of image pairs used for testing in the forest fire domain was 50 pairs, and 4000 pairs of images were cropped using a sliding window of $8 \times 10$; the number of image pairs used for testing in the TNO dataset was 100 pairs, and 8000 pairs of images were cropped using a sliding window of $8 \times 10$. All of these images maintain a uniform size of $2.6 \times 3.5$ cm, accounting for various factors such as data acquisition and processing.

To address the problems arising from the acquisition and transmission of forest fire scene images, the acquired images are preprocessed, including (1) a particle swarm optimization-based approach to grayscale forest fire scene images [14,15], based on the grayscale histogram distribution law of high pixel values in a single channel of a color image, particle swarm search is used to select the optimal threshold, as well as to automatically generate the weights of components for grayscale processing of forest fire scene images. (2) An image denoising method based on wavelet transform is used to remove the effect of noise on forest fire scene images to improve the image quality [16], the wavelet transform analysis method is established based on the idea of short-time Fourier transform localization, the image signal is segmented and refined according to multiple levels, and the wavelet coefficients obtained from the wavelet transform are appropriately processed, which can be used for suppressing the noise randomly generated by the sensor in the imaging process. (3) A frequency domain-based image enhancement method is employed to enhance the visual quality of the images, highlighting finer details in the forest fire scene images [17]. This involves processing the image signal with a specialized high-pass filter to achieve enhancement. Gaussian filtering uses a Gaussian convolution kernel to

apply different Gaussian weights to pixel gray values at various locations in the image, smoothing the forest fire scene image. (4) Based on the above data preprocessing, the SURF alignment method is used to align the visible and infrared images of the forest fire scene for the problem of non-overlap, which leads to errors in image fusion [18], extract the feature points of the forest fire scene image, use integral images to simplify the operation process, construct the Hessian matrix to extract the image feature points, construct the scale space to form the three-dimensional scale space of the feature points; obtain the direction of the feature points by calculating the harr wavelet response; form the feature description subvector based on the main direction of the feature points; add the scale factor in eliminating the erroneous feature points, choose the appropriate similarity metric to match the similar feature points, and finally obtain the completely overlapped visible and infrared images of the forest fire scene.

### 2.2. Generative Adversarial Network Fundamentals

Generative adversarial networks (GAN) were proposed by Goodfellow et al. in 2014 and consist of a generator and a discriminator [19]. The generator is the target network that aims to generate false data that match the target distribution; the discriminator is a classifier that is responsible for accurately distinguishing between real data and the fake data generated by the generator. Thus, there is an adversarial relationship between the generator and the discriminator. In other words, the generator wants to generate forged data that the discriminator cannot identify, while the discriminator wants to distinguish the forged data accurately. The generator and discriminator enhance their capabilities through continuous iterations until the discriminator can no longer differentiate between real data and the forged data generated by the generator. At this point, the generator has the ability to generate data that match the target distribution. Next, the above adversarial learning process is described formally.

Suppose the generator is denoted as G and the discriminator is denoted as D. The random data input to the generator are denoted as $Z = (z_1, z_2, \ldots, z_n)$, and the target data as $X = (x_1, x_2, \ldots, x_n) \sim P_x$. Then, the generator is committed to estimate the distribution of the target data *XP X* and generates data G(z) matching that distribution as much as possible, while the discriminator D needs to distinguish accurately between the real data X and the generated pseudo data G(z). In summary, the purpose of generating adversarial networks is to make the distribution of the faked data PG continuously approximate the target data distribution PX during continuous adversarial training. Therefore, the objective function of the GAN is defined as Equation (1):

$$\min_G \max_D E_{x_i \sim P_X}[\log(D(x_i))] + E_{z_i \sim P_z}[\log(1 - D(G(z_i)))] \tag{1}$$

In continuous iterative training, the generator and the discriminator promote each other as an adversarial relationship, continuously improving their falsification or discriminatory ability. When the distance between the two distributions becomes sufficiently small, the discriminator will no longer be able to distinguish between real and fake data. At this point, we can say that the generator has successfully estimated the distribution of the training data.

### 2.3. MMGAN Structural Design

The overall network structure consists of a generator network and a discriminator network.

#### 2.3.1. Network Structure of Generator

The multilayer convolutional structure has a powerful feature representation capability as well as a hierarchical learning capability, and with the increase in the number of convolutional layers, the model can gradually capture higher-level features as well as understand the global and local information of the image, improving the efficiency of image analysis and processing. Hence, by increasing the depth of the convolutional layers, the generator network architecture enhances the accuracy of model training. We used a

pre-trained network on ImageNet as the base network and fine-tuned it for the fusion task. For both the gradient path and the intensity path, we employed 8 convolutional layers, each utilizing a $3 \times 3$ convolutional kernel. As an example, one of these layers had dimensions $64 \times 3 \times 3 \times 128$, where 64 represents the number of input channels, $3 \times 3$ is the convolutional kernel size, and 128 is the number of output channels. Each convolutional layer includes a Leaky ReLU activation function layer and a batch normalization (BN) layer. In addition, the input of the 4th convolutional layer is feature-connected to the output of the 2nd and 3rd convolutional layers, and the input of the 6th convolutional layer is feature-connected to the output of the 4th and 5th convolutional layers. This structure helps to preserve the characteristics of the shallow network, thus effectively reducing the loss of image information. Finally, the two feature maps are interconnected along the channels to achieve full fusion of information. The step size of the last convolutional layer is set to 1, and the output uses a Tanh activation function and a $1 \times 1$ convolutional kernel to obtain the fusion result of the network. The structure of the generator network is shown in Figure 1.

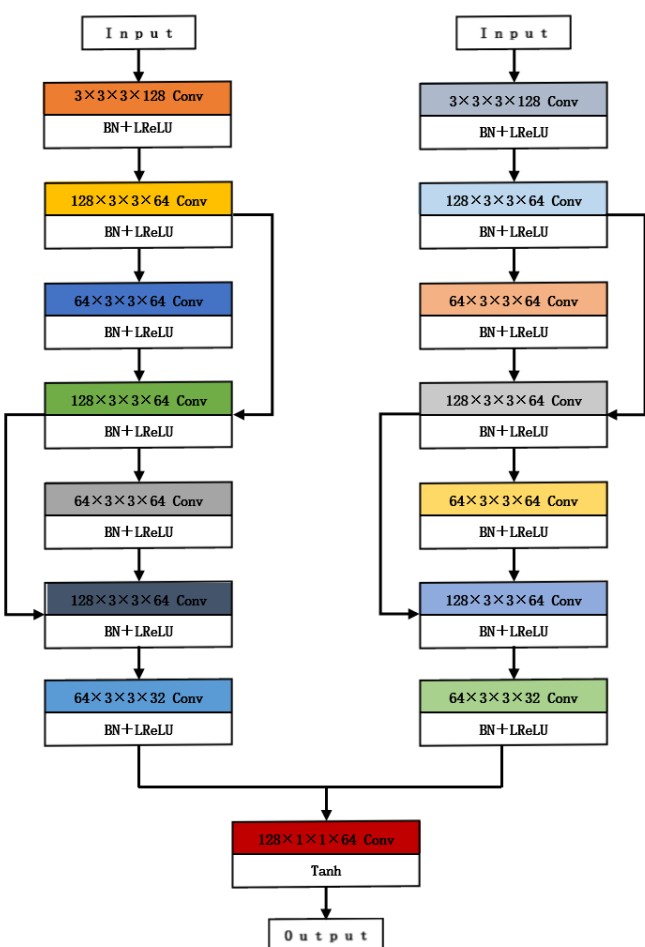

**Figure 1.** Network structure of generator.

By designing the gradient path and intensity path, the extraction of contrast information of the visible image and texture information of the infrared image is realized in the generator network structure. The gradient path aims to extract the texture information of the image, while the visible image is rich in texture detail information, so two visible images and one infrared image are coupled to one channel as the input of the gradient path; the intensity path aims to extract the contrast information of the image, while the infrared image is rich in contrast information, so two infrared images and one visible image are coupled to one channel as the input of the intensity path. This serves as the input to the

intensity path, enabling the balanced extraction of gradient and intensity information and facilitating complementary information between both images.

### 2.3.2. Network Structure of Discriminator

The discriminator is designed using a multi-level multi-classification constraint structure capable of distinguishing visible image features, infrared image features, and fused image features generated by the generator. It estimates the probability of each class within the input image, resulting in a $1 \times 2$ probability vector as output. We accomplish this by employing a pre-trained network from ImageNet as the base network and fine-tuning it for the discriminator task. The discriminator consists of four convolutional layers, and one linear layer. Four convolutional layers are used to process the input image features, all of size $3 \times 3$ convolutional kernels, all activation functions are Leaky ReLU functions, and the last three convolutional layers are set with a step size of 2 and are batch normalized by adding BN layers. The last linear layer outputs a $1 \times 2$ two-dimensional classification probability based on the image features extracted from the first four convolutional layers as input, indicating the probability of the input feature being a visible image feature $P_{vis}$ and the probability of the input feature being an infrared image feature $P_{ir}$, respectively. The generator learns a reasonable fusion strategy at that time. The discriminator network structure is shown in Figure 2.

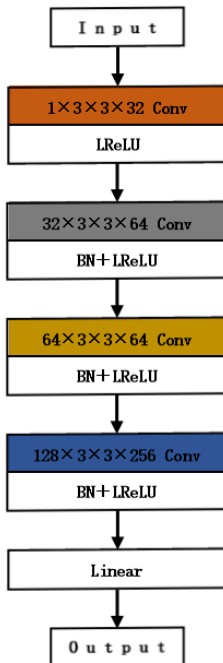

**Figure 2.** Network structure of discriminator.

### 2.3.3. General Framework of MMGAN Model

Compared with the overall GAN framework, the generator input path of MMGAN is divided into gradient as well as intensity. Firstly, for the gradient path of the generator, two visible images and one infrared image are combined along the channel dimension; secondly, for the intensity path of the generator, two infrared images and one visible image are combined along the channel dimension; then, with specific loss functions and network design, the generator can extract gradient and intensity information in a balanced way, so that the primary gradient and secondary intensity information of the visible image is collected, and the primary intensity and secondary gradient information of the infrared images complement each other. As the discriminator assesses the fused images, the generator anticipates that these images are a combination of both visible and infrared components, resulting in an expectation for the discriminator's output to yield a high probability. The discriminator's task is to accurately judge the fused image as a false image,

i.e., it expects the output probability to be both small, which excludes the possibility that the generated image is a visible and infrared image, i.e., the output image can only be a fused image. Therefore, the generator and the discriminator are continuously trained against each other. When both of the two-dimensional probabilities $P_{vis}$ and $P_{ir}$ of the discriminator output are small, the generator can independently generate information-balanced fusion images. The general framework of GAN is shown in Figure 3, and the general framework of MMGAN is shown in Figure 4.

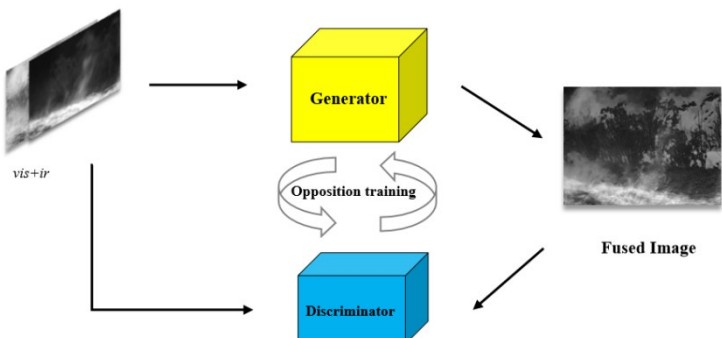

**Figure 3.** Framework of GAN model.

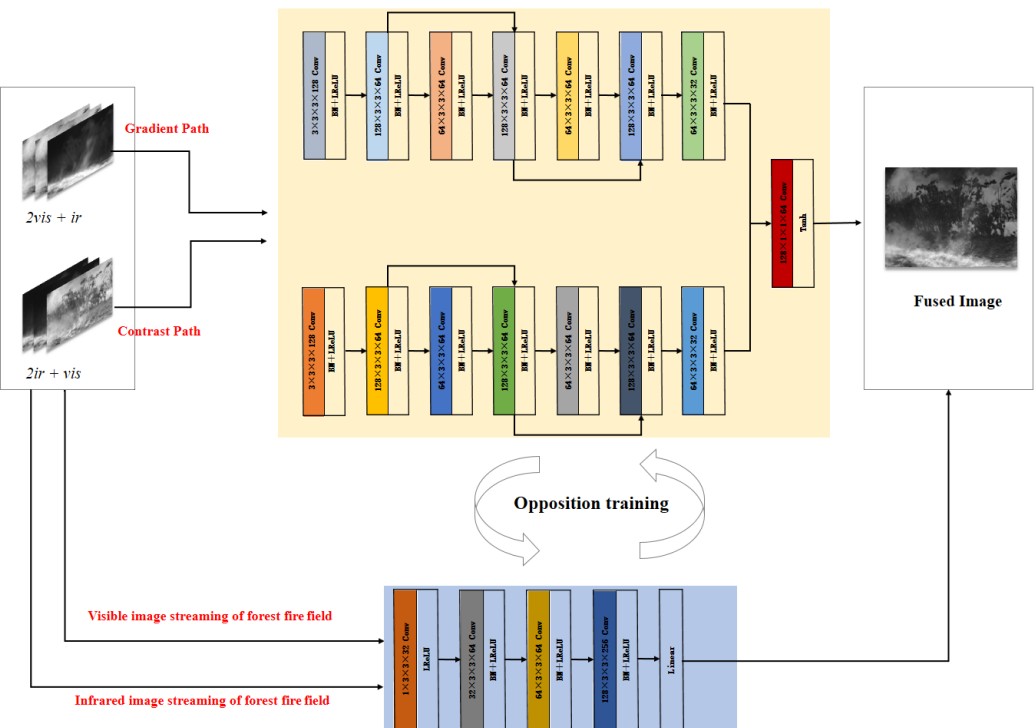

**Figure 4.** Framework of MMGAN model.

Through the aforementioned design, this paper's method can produce superior fusion results compared to traditional generative adversarial networks. It accomplishes this by preserving not only significant visible image contrast information but also by incorporating rich texture details from infrared images.

### 2.3.4. Loss Function Design

The loss functions include generator loss function $L_G$ and discriminator loss function $L_D$, where the discriminator loss function consists of visible light discriminator loss function $L_D^{vis}$ and infrared discriminator loss function $L_D^{ir}$ and forest fire discriminator loss function $L_D^{fused}$.

(1)   Generator loss function

The generator loss function consists of the generator content loss function $L_{G_{con}}$ for constraint information extraction and the generator adversarial loss function $L_{G_{adv}}$ for constraint information balancing. The specific formula is shown in Equation (2):

$$L_G = \gamma L_{G_{con}} + L_{G_{adv}} \tag{2}$$

where $\gamma$ is the regularization parameter responsible for maintaining the balance between the two items and is set to 100.

The $L_{G_{con}}$ content loss function consists of four parts: the main gradient loss, the main intensity loss, the auxiliary intensity loss and the auxiliary gradient loss, and the content loss function is shown in Equation (3):

$$\begin{aligned} L_{G_{con}} &= L_{intensity_{main}} + L_{grad_{main}} + L_{grad_{aux}} + L_{intensity_{aux}} \\ &= \delta_1 \|I_{fused} - I_{ir}\|_F^2 + \delta_2 \|\nabla I_{fused} - \nabla I_{vis}\|_F^2 + \delta_3 \|\nabla I_{fused} - \nabla I_{ir}\|_F^2 + \delta_4 \|I_{fused} - I_{vis}\|_F^2 \end{aligned} \tag{3}$$

where $\delta_1$, $\delta_2$, $\delta_3$, and $\delta_4$ are set to 1, 2, 3, and 4, respectively.

The balance between the image information can be achieved by adding the adversarial loss with discriminator to the generator's loss function, and the adversarial loss function definition $L_{G_{adv}}$ is shown in Equation (4):

$$L_{G_{adv}} = (D(I_{fused}^n)[1] - a)^2 + (D(I_{fused}^n)[2] - a)^2) \tag{4}$$

where $a$ denotes the discriminator's probabilistic labeling for the fused image, and $a$ is set to 1. The first term in the vector D(.) [1] denotes the probability that the discriminator determines that the generated fused image of the forest fire scene is a visible light image, while the second term D(.) [2] then denotes the probability that the discriminator determines that the generated forest fire fusion image is an infrared image.

(2)   Discriminant loss function

The discriminator is a multi-level multi-classifier that estimates the probability of each class of the input image. The loss function must continuously improve the discriminator discriminative power, which drives it to effectively and accurately discriminate whether the fused image is a visible image or an infrared image. Therefore, the discriminator loss function $L_D$ consists of three components to determine the loss $L_D^{vis}$ of the visible image, the loss $L_D^{ir}$ of the infrared image and the loss $L_D^{fused}$ of the fused image, and the discriminator loss function is shown in Equation (5):

$$L_D = L_D^{vis} + L_D^{ir} + L_D^{fused} \tag{5}$$

The discriminator outputs $1 \times 2$ vectors including $P_{vis} = D(x)[1]$ and $P_{ir} = D(x)[2]$.

In order to improve the discriminator's ability to recognize visible images, the predicted $P_{vis}$ needs to be close to 1 and $P_{ir}$ close to 0 when the input image is a visible image. The loss function $L_D^{vis}$ corresponding to the visible light image is shown in Equation (6):

$$L_D^{vis} = \frac{1}{2N} \sum_{i=1}^{N} \left( (P_{vis}(I_{vis}^n) - a_1)^2 + (P_{ir}(I_{vis}^n) - a_2)^2 \right) \tag{6}$$

where $a_1$ and $a_2$ are probability labels corresponding to visible light images, $a_1$ is set to 1, and $a_2$ is set to 0.

In order to improve the discriminator's ability to recognize infrared images, when the input image is an infrared image, the prediction $P_{ir}$ needs to be close to 1 and $P_{vis}$ close

to 0. The loss function $L_D^{ir}$ corresponding to the infrared image is defined as shown in Equation (7):

$$L_D^{ir} = \frac{1}{2N} \sum_{i=1}^{N} \left( (P_{vis}(I_{ir}^n) - b_1)^2 + (P_{ir}(I_{ir}^n) - b_2)^2 \right) \tag{7}$$

where $b_1$ and $b_2$ are the probability labels corresponding to the infrared images, $b_1$ is set to 0, and $b_2$ is set to 1.

Finally, in order to improve the discriminator discrimination performance, when the input image is a forest fire fusion image, both the prediction $P_{ir}$ and $P_{vis}$ need to be close to 0 in order to represent the ability of the forest fire fusion image to be neither a visible image nor an infrared image.

The loss function $L_D^{fused}$ corresponding to the fused image is defined as shown in Equation (8):

$$L_D^{fused} = \frac{1}{2N} \sum_{i=1}^{N} \left( (P_{vis}(I_{fused}^n) - c_1)^2 + (P_{ir}(I_{fused}^n) - c_2)^2 \right) \tag{8}$$

where $c_1$ and $c_2$ are the probability labels corresponding to the fused images of the forest fire, $c_1$ is set to 0, and $c_2$ is set to 0.

*2.4. Training Process and Parameter Setting*

2.4.1. Training Process

(1)  Initialize the network parameters of the generator and the discriminator, input the visible and infrared images of the forest fire scene into the corresponding discriminator, and update the parameters of the discriminator according to the output results.

(2)  Link the visible and infrared images of the forest fire scene and input them into the generator network, obtain the fused images of the forest fire scene generated by the generator, and update the parameters of the generator network by combining the corresponding generator loss. Repeat this process up to 20 times.

(3)  Input the visible image and infrared map of the forest fire scene and the fusion image generated by the generator into the corresponding discriminator, respectively, and update the parameters of the discriminator according to the output results. Repeat this process up to 5 times.

(4)  Repeat processes (2) and (3) until the network reaches equilibrium.

2.4.2. Parameter Setting

Throughout the generator and discriminator training process, we set the Leaky ReLU activation function layer parameter to 0.01, configured a batch size of 48 in the batch grouping layer, and employed Adam as the optimizer to adjust the network's weight parameters. These parameters were updated by calculating the gradient to minimize the loss function. The number of training iterations was set to 10, and the initial learning rate was set to 0.0001.

*2.5. Evaluation System*

In this paper, the performance of the fusion method is evaluated from two aspects: subjective evaluation and objective evaluation. The subjective evaluation approach relies on the observer's visual perception, and a successful fusion result should encompass both the pronounced contrast of the infrared image and the abundant texture of the visible image.

Six quantitative metrics that are widely used in the field of image fusion were chosen for the objective evaluation, including information entropy (IEN) [20], standard devia-

tion (STD) [21], average gradient (AG) [22], space frequency (SF) [23], feature mutual information (FMI) [24] and structural similarity (SSIM) [25].

$$\text{IEN} = -\sum_{l=0}^{L-1} p_l \, \log_2 \, p_l \tag{9}$$

$$\text{STD} = \sqrt{\frac{1}{M \times N} \sum_{i=1}^{M} \sum_{j=1}^{N} \left[ F(i,j) - \overline{F} \right]^2} \tag{10}$$

$$\text{AG} = \frac{1}{(M-1)(N-1)} \sum_{i=1}^{M-1} \sum_{j=1}^{N-1} \sqrt{\frac{[F(i,j) - F(i+1,j)]^2 + [F(i,j) - F(i+1,j)]^2}{2}} \tag{11}$$

$$\text{SF} = \sqrt{\text{RF}^2 + \text{CF}^2} \tag{12}$$

$$\text{FMI}_F^{AB} = \text{FMI}(A,F) + \text{FMI}(B,F) \tag{13}$$

$$\text{SSIM} = \text{SSIM}_{A,F} + \text{SSIM}_{B,F} \tag{14}$$

In Equations (9)–(14), $p_l$ is the frequency of the occurrence of points with gray value l, L is the number of gray levels of the fused image, and the larger value of STD represents the higher contrast of the image and the better quality of the image. Where, M denotes the width of the fused image, N denotes the height of the fused image, F(i,j) is the gray value of image F at pixel (i,j), F is the mean gray value, RF denotes the spatial frequency in the horizontal direction, CF denotes the spatial frequency in the vertical direction, F is the fused image, A and B are the source images, $\text{SSIM}_{A,f}$ is the brightness loss, and $\text{SSIM}_{B,f}$ is the correlation loss. The larger the value of all metrics, the better.

## 3. Results

Five image fusion methods were selected for experimental comparison with the methods in this paper: (1) anisotropic diffusion-based image fusion method (ADF) [26]; (2) guided filtering-based image fusion method (GFF) [27]; (3) convolutional neural network-based image fusion method (CNN) [28]; (4) generative adversarial network-based image fusion method (FusionGAN) [12]; and (5) image fusion method based on dual discriminator generative adversarial network (DDcGAN) [29]. The proposed methods in this paper are all pre-trained on the dataset.

### 3.1. TNO Dataset Fusion Results and Comparative Analysis

3.1.1. Subjective Assessment

To demonstrate the superiority of the MMGAN model proposed in this paper, comparative experiments were first conducted on the publicly available dataset TNO.

Figure 5 shows three sets of representative subjective evaluation results to demonstrate the performance of various methods. In the subjective evaluation comparison results, the ADF method has low contrast and does not highlight significant target features; the GFF fusion method has severe distortion, and the details in the local zoomed-in image have become illegible; the CNN method focuses too much on preserving the structural texture and neglects the preservation of the thermal radiation target; the FusionGAN method loses many texture details, and the boundary produces artifacts that make the boundary of the target widen and the whole blurred; while the DDcGAN method better maintains the saliency of the thermal radiation target in the infrared image, our proposed method excels in preserving the sharpness of the thermal target's edges and presents a superior visual effect. For example, in the first and third sets of results, the method in this paper is able to

better preserve the house structure with clearly visible edges, while other methods result in blurred contours due to edge diffusion.

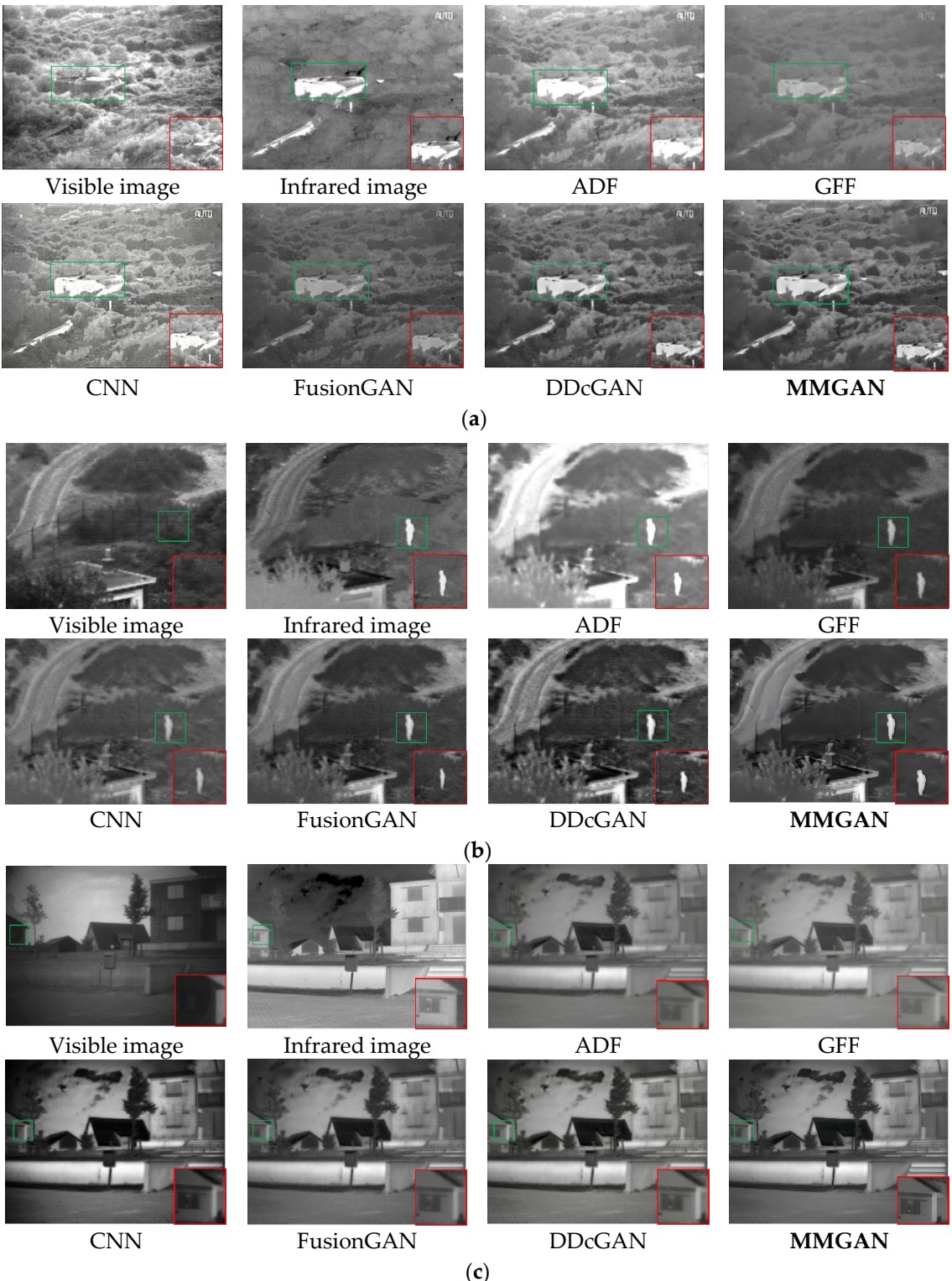

**Figure 5.** Fusion results of each model based on TNO dataset. (**a**) Group A visible and infrared fusion images; (**b**) Group B visible and infrared fusion images; (**c**) Group C visible and infrared fusion images. The red box shows an enlarged view of the green box detail.

3.1.2. Objective Assessment

In order to objectively evaluate the fusion effectiveness of ADF, GFF, CNN, Fusion-GAN, DDcGAN, and MMGAN, we conducted a quantitative analysis on 20 test im-ages from the TNO dataset, and the results are presented in Table 1. According to the data in the table, our proposed method in this paper achieved the top rank in four metrics: IEN, AG, FMI, and SSIM, and it secured the second position in STD and SF metrics, just after DDcGAN. These objective findings indicate that our method outperforms others in terms of information richness, minimal introduction of pseudo-information, high correlation with the source image, and superior contrast.

**Table 1.** Objective evaluation results for assessing the quality of fused images based on the TNO dataset.

| Metric<br>Method | IEN | STD | AG | SF | FMI | SSIM |
|---|---|---|---|---|---|---|
| ADF | 6.750 | 34.259 | 6.836 | 12.710 | 12.261 | 0.612 |
| GFF | 6.781 | 40.147 | 7.855 | 11.104 | 13.389 | 0.608 |
| CNN | 7.101 | 48.113 | 6.657 | 14.895 | 14.007 | 0.610 |
| FusionGAN | 7.431 | 51.534 | 7.472 | 15.348 | 14.099 | 0609 |
| DDcGAN | 7.642 | **52.036** | 8.059 | **17.896** | 14.325 | 0.622 |
| **MMGAN** | **8.830** | 51.265 | **9.198** | 17.764 | **14.866** | **0.656** |

Bold indicates the method of this paper; highlight bold indicates the optimal result.

Moreover, the quantitative results are in line with the visual assessment shown in Figure 5. The fusion outcomes obtained by our method exhibit a clearer and more detailed representation of the forest fire scene compared to other methods. The enhanced information richness and improved contrast in the fused images can significantly assist firefighters in comprehending the fire scene environment and making critical decisions during rescue operations.

In conclusion, our proposed method demonstrates superior performance in objective evaluation metrics on the TNO dataset compared to existing fusion techniques. We believe that the results presented in this study will have a meaningful impact on advancing forest fire image fusion techniques and contribute to more effective and safer forest firefighting efforts.

*3.2. Fusion Results and Comparative Analysis of Forest Fire Scene Datasets*

In order to further demonstrate the superiority of the MMGAN algorithm proposed in this paper, a comparative analysis of subjective and objective evaluations of forest fire scene image fusion results was performed.

3.2.1. Subjective Assessment

Three groups of typical fusion results are selected to demonstrate the performance of each method, as shown in Figure 6. Observing the fusion result plots, we can find that the ADF method contains rich texture detail information, but the noise is more obvious. The GFF has a great defect in the contrast display of the image, the texture information and contrast information in the plots are defocused, a certain number of artifacts are at the edges, and the clarity is poor. The CNN method does a good job in preserving the tree texture, but the brightness is too high, resulting in insufficient thermal radiation information. While the FusionGAN fusion method provides rich contrast information, it suffers from significant loss of infrared texture detail. DDcGAN retains more infrared texture information compared to FusionGAN, but it is slightly lacking compared to MMGAN. It is obvious from the figure that MMGAN can retain the salient targets in the IR image very accurately with almost no loss of thermal radiation intensity, and the edges remain sharp; meanwhile, it can also retain the texture details in the visible image very well. Therefore, MMGAN not only retains rich texture detail information of visible and infrared images, but also maintains

rich contrast information. Overall, the method in this paper has obvious advantages in terms of sharpness and contrast, and can achieve a good balance of visible and infrared image information.

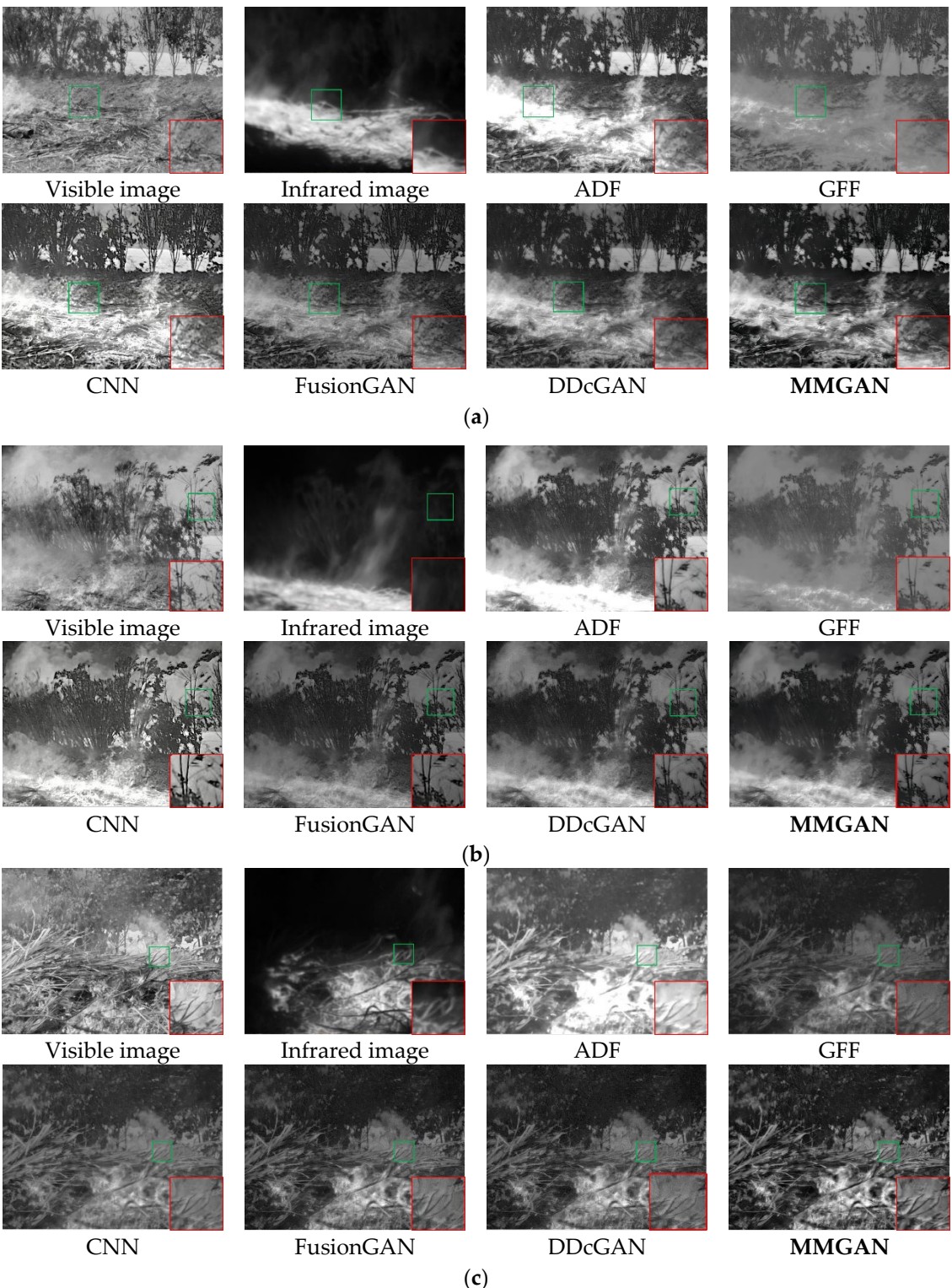

**Figure 6.** Fusion results of each model based on forest fire scene dataset. (**a**) Group A visible and infrared fusion images; (**b**) Group B visible and infrared fusion images; (**c**) Group C visible and infrared fusion images. The red box shows an enlarged view of the green box detail.

### 3.2.2. Objective Assessment

In order to objectively evaluate the fusion effects of ADF, GFF, CNN, FusionGAN, DDcGAN and MMGAN, the fusion results for forest fire scenes were quantitatively analyzed on 20 forest fire scene images, and the results are shown in Table 2. The data in the table show that the MMGAN proposed in this paper obtained the highest average value in five objective quality evaluation metrics, namely, IEN, STD, AG, FMI, and SSIM; among them, although the SF value had only the second-highest value, it can be seen that it significantly exceeded the SF value of other methods, especially FusionGAN, which indicates that MMGAN can be better than FusionGAN in maintaining a good balance of visible and infrared image information. In addition, the SF values obtained by MMGAN were much larger than those of all other methods, thus indicating that the method in this paper can better fuse the contrast and texture detail information of visible and infrared images, and the fused image information is richer.

**Table 2.** Objective evaluation results for assessing the quality of fused images based on the forest fire scene dataset.

| Method \ Metric | IEN | STD | AG | SF | FMI | SSIM |
|---|---|---|---|---|---|---|
| ADF | 6.259 | 35.744 | 7.025 | 14.259 | 12.517 | 0.612 |
| GFF | 6.753 | 35.157 | 7.724 | 15.322 | 13.27 | 0.606 |
| CNN | 6.452 | 33.51 | 7.647 | 13.865 | 12.903 | 0.664 |
| FusionGAN | 7.730 | 50.916 | 7.572 | 15.765 | 14.170 | 0.638 |
| DDcGAN | 7.849 | 52.473 | 8.983 | **17.600** | 14.836 | 0.669 |
| **MMGAN** | **8.862** | **52.674** | **9.652** | 17.347 | **14.958** | **0.680** |

Bold indicates the method of this paper; highlight bold indicates the optimal result.

In summary, MMGAN transmits the most information from the source image to the fused image, introduces the least pseudo-information, and maintains the edges best during the fusion process. The resulting fused images contain the most information, exhibit the highest contrast, and boast the richest overall texture structure. Therefore, the method in this paper is also quantitatively advantageous compared to the other fusion methods.

### 3.3. Comparative Analysis of Fusion Efficiency

Operational efficiency is one of the important metrics to assess the performance of a method. To evaluate operational efficiency, we calculated the average running times of various methods on both the TNO dataset and the forest fire scene dataset and conducted a comparison. The results are shown in Table 3. The method proposed in this paper achieves the fastest average running speed on both datasets, which is more than five times faster than other compared methods. This means that our method is able to complete the image fusion task in a shorter time, improving the overall efficiency.

**Table 3.** Average running time(s) of each model based on each dataset.

| Method | Forest Fire Scene | TNO |
|---|---|---|
| ADF | 9.476 | 7.456 |
| GFF | 5.302 | 3.259 |
| CNN | 0.562 | 0.499 |
| FusionGAN | 0.196 | 0.360 |
| DDcGAN | 0.613 | 0.613 |
| **MMGAN** | **0.143** | **0.066** |

Bold indicates the method of this paper; highlight bold indicates the optimal result.

## 4. Discussion

This study improved the model learning as well as generation capabilities by improving the design of the generator, discriminator, and loss function. Specifically, by dividing

the input path of the generator of the traditional GAN into gradient and intensity, the depth of the convolutional layer of the generator is increased to further improve the information extraction capability of the shallow network, enabling the generator to better capture the detailed features of visible and infrared images, thus improving the quality and accuracy of the fused images. At the same time, the discriminator is designed by using multiple classifiers that can discriminate the input images as visible, infrared, and fused images simultaneously and output two-dimensional probabilities, so that the discriminator improves the ability to distinguish different image types and helps to better evaluate the authenticity and quality of the fused images. Through the design of the loss function, the discriminator constrains the generator and performs inverse parameter updates, aiding the generator in gradually optimizing the quality of the generated images during continuous adversarial training. This process leads to the generation of fused images.

The study addresses the issue of potential computational slowdown due to increased model depth by modifying the network structure through the inclusion of batch normalization, residual connections, and depth-wise separable convolution. This modification reduces the number of parameters and the computational complexity of the network, enhancing the model's computational efficiency and speed. Consequently, it accelerates the training and inference processes of MMGAN, providing more timely and accurate environmental awareness and decision support for forest fire scenes.

The application of image fusion technology to the forest fire scene in our research yields significant practical benefits. By fusing images of the forest fire scene, forest firefighters can quickly understand the surrounding status of the fire, including fire expansion, smoke distribution, burning hot spots, and other key information, and can more accurately assess the danger level of the fire and make timely risk avoidance decisions based on changes in the fire environment. This is important for protecting the safety of the firefighters and improving the efficiency of fire rescue and helping reduce the casualties and property losses caused by forest fire accidents.

Current research on visible and infrared image fusion for forest fire scenes focuses on static images, while the changing situation of fires often requires the processing and analysis of continuous video data. In the future, research can delve deeper into applying visible and infrared image fusion techniques to forest fire scene video data using multi-level, multi-classification generative adversarial networks. This approach can enable more precise monitoring of fire dynamics by analyzing and fusing continuous video frames. It can also aid forest firefighters in making timely decisions and taking appropriate rescue actions, ultimately leading to a significant reduction in firefighter casualties due to forest fires.

## 5. Conclusions

To tackle the issue where traditional GANs often neglect visible contrast ratio information and detailed infrared texture details, this study introduced a multi-level and multi-classification GAN-based approach for fusing forest fire scene images from the visible and infrared spectra, and validated its accuracy. In terms of visual results, the fused images generated by MMGAN had a more significant contrast ratio and richer detailed texture information, as MMGAN could retain both the contrast ratio of the visible images and the detailed texture information of the infrared images. In terms of objective evaluation metrics, compared to the sub-optimal image fusion method, the evaluation results based on IEN, AG, FMI, and SSIM for MMGAN improved by 15.6%, 14.1%, 3.77%, and 5.47%, respectively, when applied to the TNO dataset; the evaluation results based on IEN, STD, AG, FMI, and SSIM improved by 10.23%, 0.383%, 7.46%, 0.82%, and 1.94%, respectively, when applied to the forest fire scene dataset. Furthermore, based on the same dataset, MMGAN's image fusion speed was more than five times faster than that of the sub-optimal fusion method.

In conclusion, the proposed method has better interpretability, which means it can generate more reasonable fusion strategies self-adaptively based on input images. In addition, MMGAN can retain both rich textures in detail and a significant contrast ratio of source images, effectively avoiding the weakening of beneficial information during

the fusion process. The model's computational speed is optimized while enabling better performance in the fusion of visible and infrared images. Therefore, MMGAN image fusion technology applied to forest fire scenes can quickly provide more accurate and comprehensive information on forest fire surroundings for firefighters, which helps firefighters and disaster management personnel to better understand the real-time situation inside the fire, including the location of the fire source, the degree of combustion, and the expansion of the fire, so as to provide a more accurate and comprehensive description of the fire environment. It also serves as a scientific foundation for firefighters to precisely identify and locate the fire source and other critical targets.

Finally, MMGAN image fusion technology lays a vital foundation for enhancing the intelligence and automation of forest fire scenes by integrating deep learning with image fusion technology. In the future, with the continuous development of artificial intelligence and image processing technology, MMGAN image fusion technology is expected to play an increasingly important role in forest fire rescue and disaster management, and make greater contributions to people's safety and property security.

**Author Contributions:** G.Z. and Z.Y. conceived and designed the study. Q.J. wrote the first draft and performed the data analysis. Q.J. collected all study data. S.T., H.X. and Y.W. provided critical insights in editing the manuscript. X.W. was responsible for project management. All authors have read and agreed to the published version of the manuscript.

**Funding:** This work was funded by the National Natural Science Foundation Project of China (Grant No. 32271879), the Science and Technology Innovation Platform and Talent Plan Project of Hunan Province (Grant No. 2017TP1022), the National Natural Science Foundation of China Youth Project (Grant No. 32201552) and Changsha City Natural Science Foundation (Grant No. kq2202274).

**Data Availability Statement:** The data presented in this study are available on request from the corresponding author.

**Conflicts of Interest:** The authors declare no conflict of interest.

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
