# Peer review of "Visible and Infrared Image Fusion of Forest Fire Scenes Based on Generative Adversarial Networks with Multi-Classification and Multi-Level Constraints"

_forests, doi:10.3390/f14101952_

Round 1
Reviewer 1 Report
Thanks the authors for their study. It is a valuable study, however, there are some major concerns regarding the fundamental concept behind the proposed method and its applications that need to be answered before it can be considered for acceptance.
1. The Introduction section is well written and thoroughly covers previous studies. However, more descriptions about the significance of research on visible and infrared images fusion under forest fire scenarios should be added in the introduction section.
2. One important concern with the paper centers on acquisition of the data. The description of training data in the manuscript is not sufficiently detailed. For example, from the manuscript, the reader has no way of telling whether the data was obtained from simulated forest fires or real cases, and the fuel type and geographical features of the scenarios are also not well described. Explicit and more real wildland fires are encouraged to be employed in the paper.
3. Following the previous comment, the data is augmented from 50 sets to 800 sets via cropping method, will it lead to the phenomenon of a large number of similar images appearing in the training dataset, which in turn poses the risk of overfitting? What methods are/could be implemented to address such concerns? Moreover, are evaluating samples collected in a new scenario and not included in the training dataset?
4. The authors claim that a multi-level multi-classifier was applied for discriminator. And according to the loss function of it, it seems that three classes are separated in three 1 × 2 probability vectors. Why not simply applying a single classifier with 1 × 3 probability vector and one-hot encoding to represent visible, infrared and fused image? What are the advantages of multi-classifier design?
5. It is recommended that authors present user study of MMGAN for firefighters in the result section. Except for the metrics mentioned in the manuscript, the usage experience of firefighters is also important (e.g., “can it better indicate burning area while preserving detailed fuel type features?” or “what challenging problems can it solve in the actual scenario of forest fire rescue?”). 6.And more over, which exactly improvements are important for the firefighters? Why is that? How this research can help it? It is missing.
some of the English expressions in the manuscript are somewhat confusing (e.g., “so that the primary gradient and secondary intensity information of the visible image is collected and primary intensity and secondary gradient information of infrared images and complement each other.” and “and the information extraction capability of the shallow network. capability.”).
More rigorous academic English expressions should be considered.
Reviewer 2 Report
This paper proposed a new GAN-based method MMGAN for fusing visible and infrared images of forest fire fields, which solves the problem that GANs tend to ignore visible contrast ratio information and detailed infrared texture information. Experimental results indicated that compared to the mainstream infrared and visible image fusion methods, the MMGAN model has the best visual effect and speed advantage when applied to forest fire surroundings image fusion.
The insufficiencies and areas in need of improvement in this article can be summarized as follows:
1.First and foremost, a significant issue arises in the manuscript where the authors neglect to carry out a comprehensive literature review regarding pivotal debates. Although the authors briefly touch upon the demand for forest fire field rescue and the benefits of image fusion technology within the introduction, they fail to support these claims with proper citations and in-depth discussion. It is highly advisable to incorporate research findings that demonstrate the limitations of traditional machine learning methods or alternative approaches in forest fire field rescue operations. By doing so, the authors can effectively emphasize the extent to which their work aligns with, challenges, or advances the existing body of literature. Here is a example 10.3390/rs14174362.
2.In section 2.1, it is recommended that the authors provide an illustrative introduction to the data, data enhancement, and preprocessing methods. Furthermore, the section on image enhancement and preprocessing is insufficiently detailed and would benefit from more substantial content.
3. In sections 2.3.1 and 2.3.2, it is necessary to provide theoretical or experimental evidence to demonstrate the advantages of a multi-layer convolutional structure and the process of determining the number of convolutional layers. Additionally, a brief explanation is required regarding the batch size for Batch Normalization (BN) and the parameter settings for the activation function.
4. In this paper, when introducing the loss function of MMGAN, the specific expressions for each component of the loss function are not provided. Furthermore, the training process of MMGAN is not discussed. These issues need to be addressed and corrected.
The English language quality in this paper is generally acceptable, with a decent level of grammar and vocabulary. However, there may be minor errors that require attention and revisions.
Reviewer 3 Report
The paper presents a good contribution to the body of knowledge, however it may be improved further.
Main remarks
Introduction
"In recent years, the rapid"-In this paragraph add more references at international level.
The contribution is well explained.
3.1.2. Objective assessment-Read and improve the text bettter.
"Table 2. Objective evaluation results for assessing the quality of fused images based on the forest fire field dataset"-Please replace the clour red of the numbers by bold highlighting.
Discussion-Well done.
Round 2
Reviewer 1 Report
Clearly, the comment 3 is a very important point. However, the authors do not appear to have responded it well. ‘Moreover, are evaluating samples collected in a new scenario and not included in the training dataset?’.
In addition, in the title of the manuscript, the authors claim that they are from institute 1 and 2. While institute 2 was not mentioned.
I hope the authors take the comments seriously.
Author Response
Response to Reviewer 1 Comments
Point 1: Clearly, the comment 3 is a very important point. However, the authors do not appear to have responded it well. ‘Moreover, are evaluating samples collected in a new scenario and not included in the training dataset?’.
Response 1: Thank you for your attention and questions. In this study, instead of collecting additional evaluation samples in new scenarios, the TNO public dataset and the forest fireground visible and infrared images were used as data for the evaluation and comparison experiments.The TNO dataset contains visible and infrared images from different scenarios, which were collected during field trials at three different locations, representing different military and civilian scenarios. The image pairs used for testing in the forest fire and TNO datasets are 50 and 100, respectively, while the data used for training are image blocks of 4000 and 8000 pairs obtained by cropping.
By using the TNO dataset and the visible and infrared images of the forest fire field for the evaluation and comparison experiments, we ensure the independence of the evaluation samples from the training dataset to ensure the objectivity and fairness of the evaluation results. Such an evaluation method can more accurately assess the performance and generalization ability of the model in real scenarios, and provides strong support for the reliability of the study.
Thank you again for your guidance and suggestions, and we will further revise the paper according to your comments to further improve the quality of the study.
Point 2: In addition, in the title of the manuscript, the authors claim that they are from institute 1 and 2. While institute 2 was not mentioned.
Response 2: Thank you for your feedback and careful review of the manuscript. We apologize for the oversight in not mentioning institute 2 in the title of the paper. We have now updated the title to accurately reflect the affiliations of all authors.
We appreciate your attention to detail and constructive input. Your feedback has been instrumental in improving the accuracy and completeness of our research paper.

Reviewer 2 Report
Based on the revisions made to the manuscript, it is evident that the author has effectively incorporated the feedback and suggestions provided. The author has made significant modifications to the introduction and added more details on the loss function, parameter settings, and training process.
It is clear from these revisions that the authors have made a concerted effort to address all concerns and suggestions, resulting in a more robust and well-supported manuscript.
Author Response
Response to Reviewer 2 Comments
Point 1: Based on the revisions made to the manuscript, it is evident that the author has effectively incorporated the feedback and suggestions provided. The author has made significant modifications to the introduction and added more details on the loss function, parameter settings, and training process.
It is clear from these revisions that the authors have made a concerted effort to address all concerns and suggestions, resulting in a more robust and well-supported manuscript.
Response 1: Thank you for your thorough review and positive feedback. We greatly appreciate your recognition of the efforts made to enhance the manuscript based on the feedback and suggestions provided. We are pleased that the revisions have improved the clarity and comprehensiveness of the paper.
We value your comments and suggestions and want to keep improving our research. Your feedback is vital to our work, so we welcome any kind of discussion to further improve and enhance the paper.
Round 3
Reviewer 1 Report
Thanks for the fast response.
However, in the revised version, the authors claim that 'The image pairs used for testing in the forest fire and TNO datasets are 50 and 100, respectively, and the data used for training are image blocks of 4000 and 8000 pairs obtained by cropping, respectively.' , while they previous claim 'One hundred sets of images were collected in the TNO dataset as training data. To increase the training data, the images were data enhanced and the source images were cropped using a sliding window of 8 × 10 to obtain 800 sets of visible and infrared images as the training dataset.' in the initial version. The claims are not well matched. In particular, the training data suddenly changed from 800 to 8000 while the validation is 100 which is the same as initial collected data. The elaboration of the data source is still not accurate enough.
Author Response
Response to Reviewer 1 Comments
Point 1: However, in the revised version, the authors claim that 'The image pairs used for testing in the forest fire and TNO datasets are 50 and 100, respectively, and the data used for training are image blocks of 4000 and 8000 pairs obtained by cropping, respectively.' , while they previous claim 'One hundred sets of images were collected in the TNO dataset as training data. To increase the training data, the images were data enhanced and the source images were cropped using a sliding window of 8 × 10 to obtain 800 sets of visible and infrared images as the training dataset.' in the initial version. The claims are not well matched. In particular, the training data suddenly changed from 800 to 8000 while the validation is 100 which is the same as initial collected data. The elaboration of the data source is still not accurate enough.
Response 1: Thank you for your detailed correction and review. We apologize for the inconsistencies in the description of the dataset, there were indeed errors. Below are our corrections and further clarifications on these issues:
In the original manuscript, we described 800 sets of images, when in fact we used 8000 sets of images. This error was caused by a misspelling, for which we deeply apologize. A set of images can be obtained as 80 sets of images by cropping the source images using a sliding window of 8 × 10. Therefore, 50 sets of forest fire field visible and infrared images can be obtained as 4000 sets of images by data enhancement, while 100 sets of TNO visible and infrared images can be obtained as 8000 sets of images by data enhancement.
Again, we apologize for these inaccurate descriptions, which may have been misleading. In the revised draft, we will ensure that the datasets are described more accurately and consistently to eliminate this misleading information. Thank you again for your detailed review and patience, we sincerely apologize and appreciate your help.
